# Live bearing promotes the evolution of sociality in reptiles

Ben Halliwell[1], Tobias Uller[2,3], Barbara R. Holland[4] & Geoffrey M. While [1]

Identifying factors responsible for the emergence and evolution of social complexity is an outstanding challenge in evolutionary biology. Here we report results from a phylogenetic comparative analysis of over 1000 species of squamate reptile, nearly 100 of which exhibit facultative forms of group living, including prolonged parent–offspring associations. We show that the evolution of social groupings among adults and juveniles is overwhelmingly preceded by the evolution of live birth across multiple independent origins of both traits. Furthermore, the results suggest that live bearing has facilitated the emergence of social groups that remain stable across years, similar to forms of sociality observed in other vertebrates. These results suggest that live bearing has been a fundamentally important precursor in the evolutionary origins of group living in the squamates.

[1] School of Biological Sciences, University of Tasmania, Sandy Bay TAS 7005, Australia. [2] Department of Biology, Lund University, SE-223 62 Lund, Sweden. [3] Edward Grey Institute, Department of Zoology, University of Oxford, Oxford OX1 3PS, UK. [4] School of Physical Sciences, University of Tasmania, Sandy Bay TAS 7005, Australia. Correspondence and requests for materials should be addressed to G.M.W. (email: geoffrey.while@utas.edu.au)

Social groups are extraordinarily diverse in form and function and widespread across the animal kingdom. Whereas we understand well how many social systems are maintained by selection, identifying the factors that pre-dispose certain lineages to evolve sociality in the first place remains a major challenge. Social associations between overlapping generations are a defining feature of many social systems, suggesting that delayed dispersal and parental tolerance of offspring is an important early step towards the emergence of more stable social organisation[1,2]. Understanding the evolutionary origins of social grouping, therefore, requires examination of the most basic factors that mediate social contact between parents and offspring.

Giving birth to live young and attending eggs should be fundamental in this context as these traits increase the opportunity for interactions between parents and offspring at emergence, and may therefore promote a transition from solitary to group living via kin selection[1–5]. Despite this clear prediction, empirical research investigating evolutionary correlations between live bearing, egg attendance and social grouping has been hindered by a lack of appropriate study systems. The comparative analyses necessary to address these links are not possible using traditional model systems, such as birds and mammals. First, the ubiquity of parent–offspring association in these taxa precludes any meaningful comparative analyses investigating the conditions under which social grouping initially emerged[4,6]. Second, all birds are oviparous and attend their eggs and viviparity represents a single evolutionary origin early in the mammalian radiation. Thus, the ubiquity of focal traits and evolutionary history of birds and mammals make these groups non-conducive to phylogenetically controlled tests of the relationship between parity mode, egg attendance and social grouping. To address these limitations, we need monophyletic groups that display both live bearing and egg laying, in which the phylogenetic distribution of sociality is disparate, and the expression of social associations and egg attendance are both variable and facultative.

Squamate reptiles (i.e., lizards, snakes and worm lizards) provide an outstanding opportunity to investigate these links. Firstly, facultative social associations have been reported in a diverse range of squamate species[5–7] (Fig. 1). These social associations range from transient associations between individuals to large communal aggregations with overlapping generations; however, they most commonly take the form of small family units based on delayed natal dispersal and prolonged parent–offspring associations, similar to those observed in other vertebrates[8]. Such social groups can be highly stable, with individuals maintaining social associations, including shared crevice use, communal scat deposition and regular physical contact, across multiple seasons or years[5,6,9]. Secondly, unlike mammals and birds, live bearing and egg attendance have evolved many times in different squamate lineages[10–12]. This diversity should generate considerable phylogenetic variation in the opportunity for reliable parent–offspring interactions and hence opportunities for natural selection to favour social tolerance of juveniles and, ultimately, group living.

Here, we test this hypothesis using phylogenetically controlled comparative analyses. Our results indicate considerable phylogenetic structure in the distribution of social grouping between adults and juveniles across the squamates; that the evolution of such groupings has been overwhelmingly preceded by the evolution of live birth but not egg attendance; and that live bearing is further associated with the emergence of social groups that remain stable for long periods of time (e.g., across years), comparable to the social groups of some mammals and birds. Our findings suggest that giving birth to live young promotes conditions conducive to selection for sustained parent–offspring associations and imply that live bearing has provided an important exaptation for the emergence and stabilisation of kin-based social organisation in the lineage.

## Results

**Forms of social grouping.** We examined patterns of intergenerational social grouping (henceforth social grouping), defined as the occurrence of social groups containing both adults and juveniles (see Methods for details), across 1210 squamate reptiles. We found evidence for such groupings in 95 species across 23 families (Fig. 2; Supplementary Data 1; Supplementary Fig. 1; Supplementary Table 1). Groups varied in size, form, duration, as well as whether juveniles associated with females only or with adults of both sexes. Associations between group members ranged

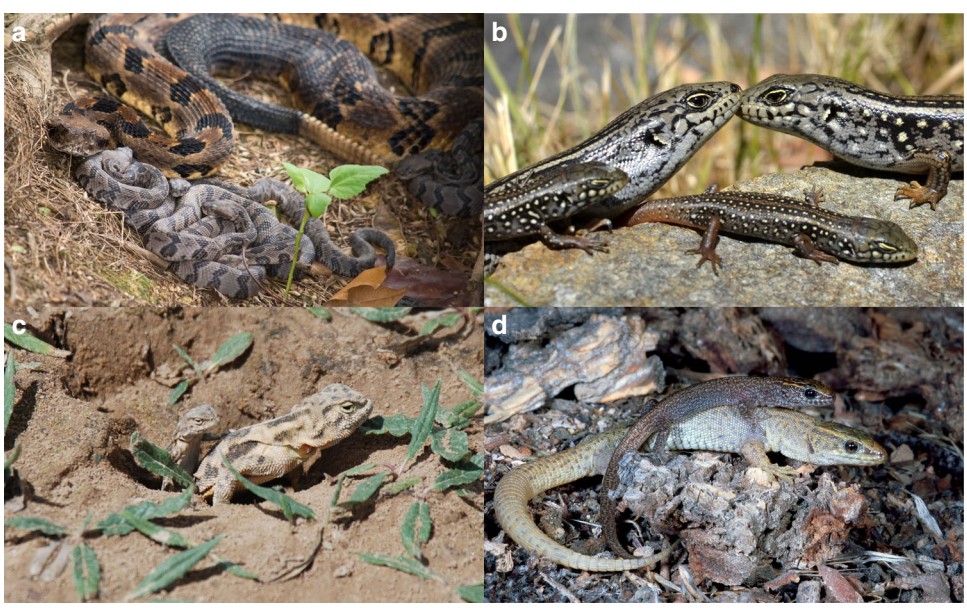

**Fig. 1** Inter-generational social grouping occurs in a diverse range of squamate reptiles. For example, social grouping occurs in the viper, *Crotalus horridus* **a**, the scincid, *Liopholis whitii* **b**, the agamid, *Phrynocephalus vlangalii* **c** and the xantusid, *Xantusia vigilis* **d**. Photos: J. Williams **a**, G. While **b**, Y. Qi **c**, A. Davis Rabosky **d**

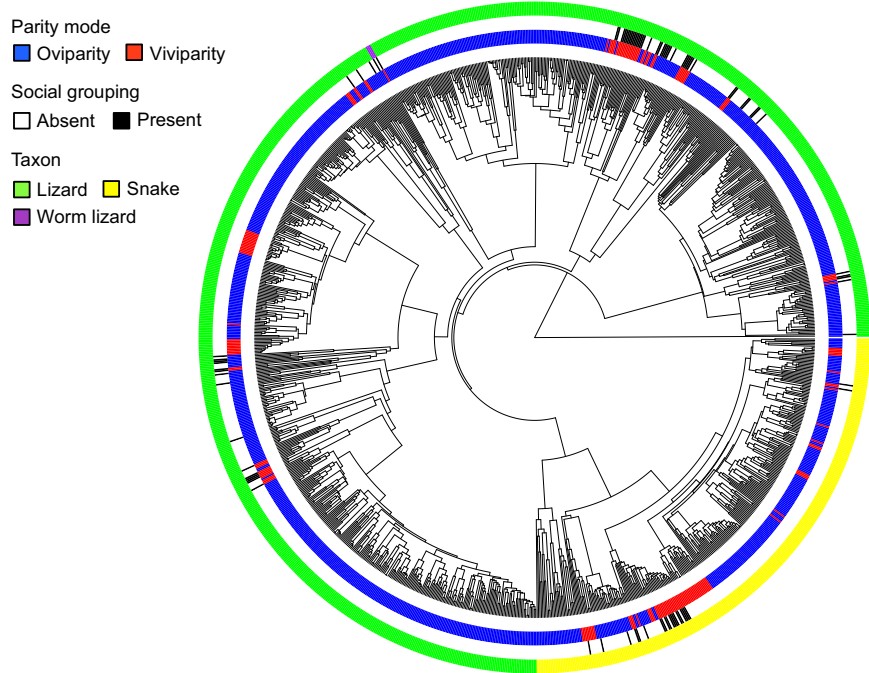

**Fig. 2** Phylogenetic distribution of social grouping across squamate reptiles. Phylogeny pruned from ref. [11] to contain only those species included in analyses (*n* = 1210, see Methods for details on species inclusion)

from passive tolerance of juveniles within adult home ranges to defense of offspring from conspecifics and predators (e.g., refs. [13–15]).

Social grouping was particularly common in Australian skinks of the Egerniinae, in which genetic studies have confirmed kin relationships among group members in 11 species to date[6]. Social associations in these species can last for several years and extend to mutual tolerance of multiple cohorts of offspring in large extended family groups[16]. Social grouping was less common and showed a more discrete phylogenetic distribution among snakes compared to lizards (Fig. 2; Supplementary Table 1). However, maternal attendance of offspring appears common in temperate pit vipers of the Crotalinae (ref. [14]; Supplementary Table 2).

**Transitions to social grouping.** We tested the extent to which the evolution of social grouping has been correlated with the evolution of both viviparity and egg attendance using a recently revised phylogeny of the squamates[17]. Social grouping occurred in viviparous species more than twice as often as in oviparous species (66 vs. 29 species), despite the overwhelming majority of squamate species being oviparous (~80% of species: ref. [11]; Fig. 2). Phylogenetic mixed modelling approaches revealed considerable phylogenetic structure in the distribution of social grouping, indicating a strong effect of tree topology, and therefore shared evolutionary history, on the presence of social grouping (Table 1). After accounting for this phylogenetic structure, parity mode remained a highly significant predictor of social grouping across species (Table 1). In contrast, among egg-laying species, social grouping was not more common in species that attend their eggs (Table 1). One important consideration for these results is the potential for the evolution of live bearing to affect speciation rates in squamate lineages[11], and therefore tree topology itself. Such non-independence of trait evolution and species diversification can bias inference from analyses that consider these processes separately[18]. However, our results were unchanged when differential rates of speciation and extinction associated with each character state were incorporated using alternative statistical

approaches[19,20] (Supplementary Table 3). Furthermore, ancestral state reconstructions supported the correlated evolution of social grouping and live bearing (Fig. 3), but not egg attendance (Supplementary Fig. 1), indicating much greater similarity between the reconstructed histories of transitions toward viviparity and toward social grouping than expected by chance (Fig. 3b). Indeed, while social groups containing adults and juveniles have arisen independently in both viviparous and oviparous lineages, transitions toward a state of social grouping have occurred at a considerably higher rate in viviparous lineages (Supplementary Table 3), implying that viviparity more often precedes the evolution of group formation. This result holds true whether or not transitions in parity mode are bi-directional or constrained to be irreversible (i.e., oviparity to viviparity; Supplementary Table 3). We found no support for the alternative causal explanation that social grouping promotes the evolutionary emergence of viviparity (see Methods for details).

**Transitions to stable social grouping.** Some squamates are known to exhibit stable social grouping, defined as social groups that are consistently maintained or those in which individuals maintain group membership across multiple seasons or years (see Methods for details). These are the types of social grouping most likely to rely on close kin and hence their evolution should be particularly likely in viviparous lineages. Our literature review revealed 21 species reported as displaying this form of sociality (Supplementary Table 1; Supplementary Data 2), 18 of which featured in our phylogeny[17] and could therefore be included in analyses. Despite the limited sample size, phylogenetic mixed modelling indicated strong positive effects of viviparity on the occurrence of stable social grouping (parity: $Z = 3.23$, $P = 0.001$, citation count: $Z = 0.48$, $P = 0.633$). Indeed, of the 18 species that exhibited stable social groupings, 17 (94%) were viviparous. Ancestral state reconstructions once again suggested a strong evolutionary correlation, indicating that live birth preceded the evolution of stable social organisation in 13 of the 14 independent origins of this trait (Supplementary Figs. 2, 3).

**Table 1 Phylogenetic generalised linear mixed models (PGLMM) testing the influence of parity mode and egg attendance on the occurrence of social grouping in squamate reptiles**

| Data set | Parameter | Estimate | Test statistic |
|---|---|---|---|
| *Parity mode* | | | |
| Conservative (*n* = 324) | Intercept (β0) | −2.01 ± 0.96 | **Z = −2.09, P = 0.04** |
| | Parity mode (β1) | 2.34 ± 0.42 | **Z = 5.59, P < 0.001** |
| | Citation count | <0.001 ± 0.001 | Z = 0.27, P = 0.79 |
| | Signal in residuals (s2) | 2.84 | **P < 0.001** |
| | Signal in response (s2) | 4.19 | **P < 0.001** |
| Relaxed (*n* = 1210) | Intercept (β0) | −3.25 ± 1.50 | **Z = −2.16, P = 0.03** |
| | Parity mode (β1) | 2.82 ± 0.49 | **Z = 5.70, P < 0.001** |
| | Signal in residuals (s2) | 7.28 | **P < 0.001** |
| | Signal in response (s2) | 8.92 | **P < 0.001** |
| *Egg attendance (analyses restricted to oviparous species)* | | | |
| Conservative (*n* = 219) | Intercept (β0) | −1.61 ± 0.99 | Z = 1.62, P = 0.11 |
| | Egg attendance (β1) | −0.83 ± 0.54 | Z = −1.52, P = 0.13 |
| | Citation count | <0.001 ± 0.001 | Z = −0.06, P = 0.95 |
| | Signal in residuals (s2) | 2.66 | **P < 0.001** |
| | Signal in response (s2) | 2.35 | **P < 0.001** |
| Relaxed (*n* = 1049) | Intercept (β0) | −3.89 ± 1.51 | **Z = 2.58, P = 0.01** |
| | Egg attendance (β1) | 1.04 ± 0.63 | Z = 1.66, P = 0.10 |
| | Signal in residuals (s2) | 7.07 | **P < 0.001** |
| | Signal in response (s2) | 7.92 | **P < 0.001** |

*N equals the number of species included in each analysis*
*Significant terms are shown in bold*
*The 'signal in response (s2)' parameter is derived from a model fit with no predictor variables and provides an estimate of phylogenetic structure of social grouping from each data set. Model estimates are reported ± SE. For confidence intervals of parameter estimates, see Supplementary Table 4. For a details on the conservative and relaxed data set, see Methods*

## Discussion

Our results show that the evolution of social group formation in squamates has been overwhelmingly preceded by the evolution of live birth across multiple independent origins of both traits. Furthermore, live bearing is associated with the emergence of permanent social groups or groups in which individuals maintain membership across multiple seasons or years, many of which represent kin-based sociality similar to that seen in mammals and birds.

How does the evolution of live birth facilitate the emergence of social grouping and social complexity? The most parsimonious explanation is that live birth results in a higher recurrence of physical association between parents and neonates, allowing for more consistent selection on social interactions among related individuals[1,2,5,21] (see also ref. [22]). This may be particularly important in squamate reptiles which, in contrast to many birds and mammals, have relatively low costs of tolerating juveniles[23,24]. Therefore, these costs may easily be outweighed by benefits when juveniles residing within adult territories are likely to be kin, mortality during dispersal is high, and low habitat availability restricts successful settlement (see below). While the costs of tolerating juveniles should be similar in egg-laying species, the lack of close physical association between parents and neonates at emergence may mean that the opportunities for selection to act on kin interactions are constrained, even if habitat availability restricts successful settlement away from the hatching site. The one exception to this is in oviparous species in which females also attend their eggs. Interestingly, we found that egg attendance did not pre-dispose social associations between adults and juveniles in oviparous species. However, as egg attendance can occur at any time during the incubation period and we are unable to assign exact timing of egg attendance with respect to hatching in most instances based on current data, this variable may be a relatively poor predictor of physical association between mothers and offspring compared to live birth. Alternatively, live birth may promote selection on social traits via mechanisms other than post-partum interactions, e.g., by facilitating kin recognition via prolonged physiological exchange between mothers and offspring throughout gestation[25–27]. Indeed, many viviparous lizards and snakes are capable of kin recognition[28–30], even after experimental separation from mothers and siblings at birth[31–33], whereas experimental demonstration of kin recognition is lacking in oviparous species (but see ref. [34]).

Irrespective of the mechanism linking viviparity to increased parent–offspring association, the subsequent localisation of offspring within parental territories is an important first step towards more complex forms of social behaviour and parental care[2,35]. Indeed, our analyses suggest that live birth also preceded the evolution of more stable social organisation. This result conforms well to the observation that 17 of the 18 lizard species in the phylogeny thought to display stable, kin-based family groups are viviparous[5,6], and that all species in our data set for which kin-based sociality was confirmed via molecular methods were also viviparous (Supplementary Data 2). Thus, by promoting repeated social contact between generations, live birth may set the stage for a gradual evolution towards more stable forms of social organisation (such as family living) as well as more complex parental care, including defense of offspring after birth[35,36].

Despite the strong relationship between live bearing and social grouping, the relative rarity of social groups among squamates, as well as its occurrence in oviparous species, indicates that viviparity alone is insufficient for social groups to form. Indeed, the emergence of social grouping depends not only on an increased opportunity for social interaction but also on the fitness benefits from those interactions[2,37,38]. Research into the evolution of family living in other systems suggests that ecological and life history characteristics act in concert to bias the cost-benefit trade-off of delayed dispersal and create demographic conditions conducive to philopatry and, thus, cooperative behaviour among kin[39–41]. Similar processes could be driving the expression of social grouping among the squamates. For example, life history characteristics that favour natal philopatry by limiting the turnover of habitat vacancies in the local environment, such as slow maturation, long life-span and high adult survivorship, combined with habitat characteristics that promote social contact through individual reliance on spatially aggregated resources, could facilitate the

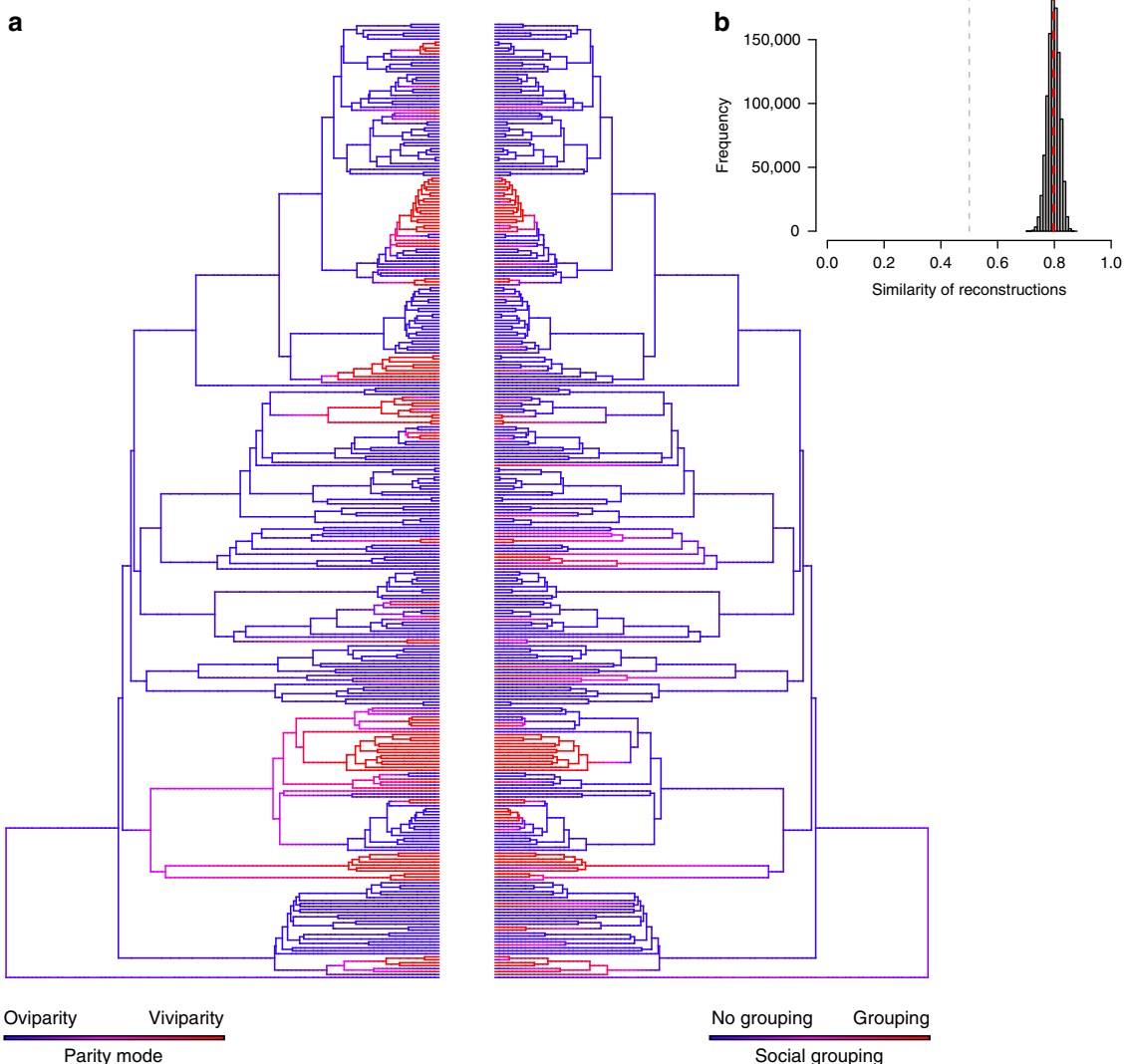

**Fig. 3** Correlated evolution of viviparity and social grouping among squamate reptiles. **a** Ancestral state reconstructions of parity mode and social grouping by stochastic character mapping. Phylogeny restricted to species in the 'conservative' data set ($n = 324$, see Methods for details). Branch colours represent posterior probability densities of edge states based on 1000 stochastic character maps of each reconstruction. **b** Distribution of similarity scores between stochastic character map sets based on separate ancestral character state reconstructions of parity mode and social grouping. The grey line represents the null expectation of similarity between map sets assuming parity mode, and social grouping shows no evolutionarily correlation during reconstruction[65]. The red line represents the mean similarity between map sets based on our reconstructions

emergence of social grouping (see ref. [7] for discussion). Indeed, all squamate species studied to date that have been shown to live in stable social aggregations have an ecology and life history that is conducive to the emergence of family living[7]. Once stabilised, the occurrence of delayed dispersal then provides the social context for the evolutionary elaboration of cooperative behaviour[1,42], expedited by kin recognition mechanisms that facilitate selection on beneficial social interactions between parents and offspring[3,43–45]. These arguments generate testable predictions that offer exciting opportunities for future comparative and empirical studies.

In conclusion, our comparative analyses reveal that social groups involving adults and juveniles have evolved multiple times in squamate reptiles. Live bearing, but not egg attendance, appears to facilitate the emergence of social groupings, suggesting that giving birth to live young promotes conditions conducive to selection for sustained parent–offspring association. These results imply that live bearing has provided an important exaptation for the emergence and stabilisation of kin-based social organisation in lizards and snakes.

## Methods

**Data collection**. We conducted two extensive literature searches to generate the data used in this study: one very broad search focusing on reports of parental care, including both pre- and post-partum care behaviours, and another more targeted search focusing on social grouping. Our reasoning for conducting a general literature search on parental care traits was three-fold: (1) to review and collate all available data on squamate parental care for use in future research projects, (2) to collect data on egg attendance behaviour for use in our phylogenetic comparative analyses and (3) to uncover additional reports of social grouping between parents and offspring not revealed by search terms specific to sociality.

We began our search of parental care traits by extracting all relevant data from the most comprehensive review of parental behaviour in squamate reptiles to date[46]. We augmented these data, excluding any reports that were highlighted in the text as uncertain, then extended upon it by adding new reports of parental care and updating previous reports based on current literature. We used ISI Web of Science to search for all articles from 2003 to the present (November 2016) using the search terms 'reptile', 'lizard', 'snake', 'amphisbaenia' and 'squamate', combined with 'nest*', 'guard*', 'defense', 'brood*', 'provision*', 'parental', 'care', 'tolerance' and 'oophagy'.

Our literature search of parental care yielded many reports of parent–offspring associations in which offspring were observed to delay natal dispersal and remain in close proximity to parents after hatching or birth (Supplementary Tables 2, 4; Supplementary Data 1), forming inter-generational social groups. We singled out

these reports as evidence of social grouping and combined them with those from a recent review of social aggregation in squamate reptiles[6], including only those species reported to display aggregations containing both adults and juveniles. We then used ISI Web of Science to perform an independent literature search using the search terms 'reptile', 'lizard', 'snake', 'amphisbaenia' and 'squamate', combined with the terms 'natal/offspring dispers*', 'birth*', 'birth site', 'territor*', 'kin', 'social system', 'social interaction', 'post hatching', 'post birth' and 'reproductive ecology', both to reveal additional reports of social grouping and provide supporting evidence for assigning absence of social grouping within a species (see 'Assigning absence of social grouping' in the Methods). We treated social grouping as a discrete variable, coding it as 'present' based on reports of aggregations containing both adults and juveniles (including parent–offspring associations that persist beyond the birthing/hatching period), and 'absent' if substantial or targeted literature had failed to report presence of the trait (see 'Assigning absence of social grouping' in the Methods). Wherever possible we accessed the primary literature cited in reviews to confirm reports of these associations, but relied on the interpretation of authors when we could not access primary sources. We also collected data on the presence of stable social grouping in squamate reptiles. We followed the definition of stable social grouping set out in ref. [6], defined as (1) permanent aggregations, i.e., social groups are maintained all throughout the year or (2) where aggregations are periodic or seasonal, social groups in which individuals maintain group membership across multiple seasons or years. We collected all data on the parity mode of species from supplementary materials in ref. [5]. We did not differentiate between viviparity and ovoviviparity, and excluded from analyses species that were reported as displaying both oviparity and viviparity (e.g., *Zootoca vivipara*).

### Quantifying the functional and taxonomic diversity of parental care.

In order to summarise the diversity and distribution of care traits across the squamate lineage, assign absence of social grouping based on the presence of other care behaviours (see 'Assigning absence of social grouping' in the Methods), as well as identify oviparous species displaying 'egg attendance', we organised similar forms of parental behaviour into five broad categories that encapsulate many primary forms of care displayed by squamate reptiles (Supplementary Table 5; Supplementary Data 1). These were nesting behaviour, egg manipulation, defense, neonatal assistance and post-partum parent–offspring associations.

For nesting behaviour, we only included reports where females directly and physically manipulated substrates into an egg-receiving site as nesting behaviour. We did not include reports of mothers displaying selectivity for particular egg-laying site characteristics, as some degree of selectivity in site selection was considered to be ubiquitous across oviparous squamates. We defined a species as displaying parent–offspring associations if offspring remained in social contact with a parent (typically the mother) beyond the time of hatching or parturition. To be assigned parent–offspring association, reports of overlap between adult and juvenile space use needed to be corroborated with observations of interactions and/or tolerance between adults and juveniles (e.g., juveniles in direct contact with an adult, adults and juveniles basking in close proximity, juveniles and adults found sharing crevice sites, burrows etc.). Oviparous species that displayed 'egg manipulation', 'defense' or 'neonatal assistance' (Supplementary Table 5) were considered to display 'egg attendance' in all relevant analyses.

### Assigning social grouping.

The main aim of our review was to detect reports of social grouping between adults and juveniles (SG) indicative of tolerant social interactions among overlapping generations. Therefore we accepted reports of adults grouping together with 'juveniles', 'hatchlings', 'neonates', 'offspring' and 'young', but not 'sub-adults' or 'yearlings'. As it is so rarely reported, we did not impose a specific distance between adults and juveniles necessary to demonstrate social grouping, but instead relied on author's interpretations of grouping behaviour, e.g., species with descriptions of adults and juveniles forming long- or short-term associations (based on delayed post-natal dispersal), sharing crevice or refuge sites, seen 'basking together', or described as forming 'aggregations' or 'groups', were considered to display social grouping. To confirm reports, we conducted individual searches on all species reported to show parent–offspring association (Supplementary Table 5) or other grouping behaviour between adults and juveniles, entering the species name (including recent taxonomic revisions) as the search term and critically reviewing any relevant literature. In some cases, particularly within the review of ref. [14] of parental care in vipers, reports of SG for some species have been compiled from personal communications with field herpetologists rather than from discrete published sources. Although lacking primary literature, these reports represent detailed observations from trained herpetologists and in many cases have been confirmed by subsequent studies (e.g., refs. [47–49]) and were therefore retained.

### Assigning absence of social grouping.

Phylogenetic analyses of character state evolution inevitably suffer from incomplete data sets (see refs. [50,51] for a discussion of the difficulties associated with missing data). The challenge of missing data also extends to the assignment of true negatives. For example, aside from general statements about the paucity of sociality across the reptilia, the absence of SG in any given species is almost never explicitly stated, resulting in potential biases being introduced into phylogenetic reconstructions[52]. Various methods have been

proposed for reducing the influence of such missing data on model estimates (reviewed in ref. [53]). However, depending on the availability and distribution of data points across the phylogeny, different methods can produce highly divergent model estimates[53] and a cautious approach is advised for any phylogenetic analyses with incomplete data coverage[54]. Therefore, considering the limits of current data, we chose to address incomplete data coverage and assignment of the absence of social grouping in two ways. First, we used strict criteria based on comprehensive literature searches to assign absence of SG (see below). Second, we repeated each analysis on two separate data sets in which different methods of assigning zeros were applied (see 'Statistical analyses' in the Methods section) and checked for qualitative consistency between model outputs.

During our literature search of both parental care and social grouping (Methods), we read each publication title returned by our search terms and critically accessed all relevant articles. Within each relevant article we scanned the abstract and methods sections to determine field or laboratory methods and searched for the terms 'young', 'offspring', 'juvenile', 'neonate', 'social', 'aggregat', 'care', 'birth', 'hatch', 'parental' within the text. We scanned all paragraphs containing these terms and if an article was suspected to contain relevant information based on these searches, we read it in full.

In some cases, authors explicitly stated that juveniles did not associate with adults (e.g., refs. [55–57]), or that mothers abandoned eggs or neonates immediately after oviposition or parturition (e.g., refs. [14,58]). However in the majority of cases, the absence of parent–offspring or adult–juvenile associations was not explicitly stated or could not be inferred directly from the text. Therefore we assigned SG as being absent in a species if it met one of the following three criteria: (1) Behavioural forms of parental care other than parent–offspring association had been reported (Supplementary Table 5), based on aforementioned literature searches, with no mention of associations between adults and juveniles in any of the literature accessed. (2) Studies of life history, reproductive ecology, spatial ecology or habitat use were available in which researchers conducted observations and/or field collections during periods of hatching or parturition, collected observations of both adults and juveniles, but did not report any social association between these age classes. We further refined this latter criterion by only including studies that used manual methods of animal capture (i.e., we excluded studies relying solely on passive trapping) to ensure researchers had considerable opportunities for observing associations between adults and juveniles. (3) The species is considered to be well studied, defined as having a citation count of ≥100 peer-reviewed publications, making it highly unlikely that such behaviour would not have been reported if present. Our threshold of 100 citations is conservative given the methods used in recent comparable studies (e.g., refs. [59,60]). However, more importantly, we incorporated information on the reproductive biology of species whenever available to inform decisions regarding the coding of absence of social grouping, and our use of multiple exclusion criteria (above) provides supporting evidence for most species, as only 20 species were assigned absence of social grouping based solely on having >100 citations.

To conduct these searches, we used Scopus to search article titles, abstracts and keywords, using species names as the search term. We used Scopus for these searches instead of ISI Web of Science because the KeywordsPlus function of ISI Web of Science inflated estimates of the number of articles directly relating to the species in question.

### Statistical analyses.

We used three different methods to test the hypothesis that social grouping has evolved more readily from viviparity than from oviparity; phylogenetic generalised linear mixed modelling (PGLMM), multi-state speciation and extinction (MuSSE) modelling and ancestral state reconstructions by stochastic character mapping.

In the first method, we used binary PGLMMs[61–63] implemented in the R package 'ape' to provide coefficient estimates and statistical tests of each predictor in our model while accounting for phylogenetic signal in the response variable. Fit without predictors, these models provide a test for phylogenetic signal ($s2$) in the response[64]. Specifically, an $s2$ value of 0 implies no phylogenetic signal and therefore that the distribution of the trait of interest is random with respect to tree topology, i.e., is not influenced by the phylogenetic relationships between species. Increasing $s2$ values imply that an increasing proportion of variance in the distribution of the trait is explained by tree topology. Thus, we first fit a model with no predictor variables to test for phylogenetic structure in the distribution of social grouping across the tree (Table 1: 'signal in response' values). We then fit a model including parity mode and citation count as predictors to confirm the influence of parity mode on the occurrence of social grouping after accounting for phylogenetic structure and that our data were not biased by available literature on each species, respectively (Table 1). Finally, to test the hypothesis that egg attendance also facilitates social grouping among oviparous species, we repeated this procedure fitting a model with egg attendance as the predictor variable in analyses restricted to oviparous species (Table 1).

For each of these models, we used parametric bootstrapping to evaluate uncertainty in the intercept ($\beta_0$), main predictor ($\beta_1$) and phylogenetic signal ($s2$) estimated from each PGLMM (Supplementary Fig. 4). This was achieved by an iterative simulation procedure. We first simulated the evolution of each predictor variable (parity and egg attendance) across the phylogeny, specifying the rate of character change as that estimated by a two-state Markov model of ancestral

character estimation. Using these simulated predictor values and the parameter estimates from our original PGLMM fit, we simulated response data (i.e., presence absence of social grouping) and re-fit the model with these simulated predictor and response values. We repeated this procedure 1000 times for the conservative data sets and 500 times for the relaxed data sets (see below for details; reduced number of simulations due to computational limitations) then checked each original parameter estimate against the distribution of estimates for that parameter returned from the simulations (Supplementary Fig. 4). This allowed us to evaluate both bias and variance in the parameter estimates returned by our models and confirm that 99% of the distribution of simulated estimates for both $\beta_1$ and s2 did not cross zero (Table 1; Supplementary Fig. 4).

Second, we used MuSSE models[19,20], implemented in the R package 'diversitree'. MuSSE models provide estimates of transition rates between different character states while accounting for variation in speciation and extinction rates associated with each state[19]. Accounting for such variation is potentially important in our context, as viviparity has been associated with comparatively high rates of speciation within the squamates[11] and may therefore lead to biased inference from analyses that assume independence between rates of trait evolution and patterns of species divergence (i.e., PGLMM). We first combined our two binary characters (parity mode: oviparous, viviparous; and social grouping: present, absent) into four unique character states (Supplementary Fig. 5), and then estimated transition rates between each character state based on maximum likelihood reconstructions (see ref. [19]). We began by fitting a model that allowed transition rates from a state of 'social grouping absent' to a state of 'social grouping present' to vary depending on parity mode. We then fit a constrained version of the model in which these transition rates were forced to be equal, and compared model fits using a likelihood ratio test ('Fit' vs. 'Test' models in Supplementary Table 3). If the model allowing transition rates to vary is favoured by AIC, this test indicates that transitions to social grouping have occurred at different rates in oviparous and viviparous lineages.

Third, we performed ancestral state reconstructions via stochastic character mapping in the R package 'phytools'[65]. We chose stochastic character mapping over marginal or joint likelihood approaches because this method allows for more robust estimation of error associated with parameter estimates by incorporating MCMC sampling. As analyses were performed on a maximum likelihood tree[17], we were not able to incorporate phylogenetic uncertainty into reconstructions by sampling from a posterior distribution of trees. Therefore we used an empirical Bayesian approach, in which we fit a continuous time reversible Markov model for the evolution of each of our two binary traits (parity mode and social grouping) to the tree, and then simulated 1000 stochastic character histories for each reconstruction using the model fit and tip states (see ref. [66]). This allowed us to estimate transition rates between states as well as the number of independent origins of social grouping from the posterior distribution of parameter estimates from each map set. Finally, we used this approach to estimate the strength of correlated evolution between viviparity and social grouping by calculating the similarity of reconstructions between map sets[65] (Fig. 3b).

We repeated the above procedure, using each statistical method described (PGLMM, MuSSE and stochastic character mapping) to investigate the evolutionary correlation between egg attendance and social grouping among oviparous species. We also used PGLMM and stochastic character mapping to investigate the evolutionary correlation between parity mode and stable social grouping. Due to limited data on stable social grouping, these analyses were only performed on the 'conservative' data set (see below). We used a recently published time-calibrated squamate phylogeny[11,17], including the tuatara, *Sphenodon punctatus*, as an outgroup in all analyses. Due to controversy over whether reversals from viviparity to oviparity are biologically plausible[25,52,67–70], we fit MuSSE models both allowing and prohibiting transitions from viviparity to oviparity and report the output of all models (Supplementary Table 1). Additionally, because multiple transitions in adaptive traits are unlikely to arise simultaneously[71], simultaneous double transitions (i.e., from a state of oviparity without social grouping to a state of viviparity with social grouping) were also prohibited (Supplementary Fig. 5).

Finally, due to challenges arising from incomplete data coverage[40,41], we also performed PGLMM and MuSSE analyses on two alterative data sets, representing a conservative and relaxed approach of including species, to confirm qualitative consistency of results. To construct the 'conservative' data set, all species for which reliable reports of social grouping were available and which were included in the phylogeny, we coded as '1' ($n = 85$) and species meeting criteria for absence were coded as '0' ($n = 239$). For the 'relaxed' data set, we relaxed the criteria for assigning absence of social grouping, allowing us to incorporate more species into these analyses. Specifically, in addition to species included in the 'conservative' data set, we added species, coded as '0' for social grouping, if they belonged to a taxonomic family for which no reports of social grouping were found for any species representing that taxonomic family during our literature search ($n = 886$). We chose complete absence of reports of social grouping at the family level because this was the highest taxonomic classification possible within the order squamata and therefore the most conservative approach for assigning zeros based on an 'absence of reports' method.

**Alternative causation**. Taken together, our results strongly suggest a causal connection between viviparity and the evolutionary emergence of social grouping, however they do not preclude alternative causal explanations entirely. We fit additional MuSSE models to explore the possibility that social grouping predisposes the evolution of viviparity, but found no support for this alternative causal explanation (Supplementary Table 6). Furthermore, we know of no biologically plausible mechanism by which the occurrence of social grouping may promote or facilitate transitions to live bearing, whereas strong arguments exist for the opposing causal pathway[4–6].

**Data availability**. All data generated or analysed during this study are included within the paper and its Supplementary Information files.

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

## Acknowledgements

This work was supported by the Australian Research Council (DP150102900; awarded to G.M.W. and T.U.) and an Australian Bicentennial Scholarship (awarded to B.H.). T.U. was supported by the Royal Society of London and the Knut and Alice Wallenberg Foundation. G.W.M. was supported by an ARC DECRA fellowship (DE150100336). We thank Louis A. Somma for correspondence during manuscript preparation. We thank Charlie Cornwallis, Simon Blomberg and Chris R. Cooney for feedback on statistical methods.

## Author contributions

G.M.W., B.H. and T.U. conceived of the study. B.H. conducted literature searches, data collation and statistical analyses. B.R.H. provided consultation on statistical approaches and contributed to statistical analyses. B.H., G.M.W. and T.U. prepared the manuscript with assistance from B.R.H.

## Additional information

**Competing interests:** The authors declare no competing financial interests.

