## [Peer Review File · Nature Communications]

Reviewers' comments:

Reviewer #1 (Remarks to the Author):

Evolution of sociality is subject of much interest and reptiles so far were quite poorly represented in phylogenetic analyses of sociality. A major strength of this paper that it takes up this challenge and using impressive dataset and methodology, proposes a link between live bearing and sociality.

The topic is of general interest, the analyses are professionally done and the writing-up is excellent.

I have 4 comments.

1. Data extraction was focused on traits related to parental care eg brood defence, provisioning and parental care. These traits are only a small subset of "sociality", so I'm confused why the focus of the paper is on sociality rather than on care. Social grouping just confuses this issue, and it is never made clear what's the difference between sociality and "stable social grouping".

For a non-herpetologist all of these traits represent various aspects of caring, and there is a bit of leap in logic to sociality.

Reptiles could be considered having some sort of sociality, although this seems strikingly different from the sociality seen in fishes, birds and mammals.

2. Numerous species that were scored "social" seem to exhibit a simple trait: egg-manipulation. It is a bit fuzzy what this trait means and even less clear whether this behaviour would classify parental care at all.

3. Reporting bias is an issue in understudied groups like amphibians and reptiles. I wonder whether the cut-off line (>100 publications) is the best way of filtering out pseudo-absences in traits. For example, a species can be very well studied from taxonomic point of view although there may be still no detailed study on its breeding system so that the traits of interest could be easily missed.

4. Science is moving toward data sharing and open access, and I find it weird that the authors do not intend making their data publicly available. This is not acceptable and would prevent others from checking the data and potentially following up the study.

The dataset and their sources should be made available in open access data depository like Dryad.

Reviewer #2 (Remarks to the Author):

In this paper, the authors assess the evolution of sociality across squamate reptile. In addition to providing a further, detailed description of variability in sociality across this thus far understudied group of animals, their results indicate that giving birth to live young strongly increases the chance that groups with overlapping generations form. This manuscript adds to a growing literature of comparative studies that indicate that sociality tends to be rare, and generates insight that are not only relevant for this particular clade but for the evolution of sociality in animals more generally.

My problem with the current version of the manuscript is that, for me, this very short manuscript leaves out crucial information both for people not familiar with the subject and those potentially wanting to build on this study. Adding further detail will help readers to understand definitions and how they compare to other taxonomic groups, be able to replicate their approaches, and clarify potential causality underlying the detected links. This is an impressive data-set (though I have not checked the actual classification of species as I am not familiar with this group of animals), the analyses appear solid, and the authors are careful in not drawing speculative conclusions. My comments are mainly to provide additional clarifications and information that might help readers to better follow the arguments.

Definitions:

Neither the main text nor the method contain a sufficient definition of "sociality" that would allow me to go to the primary literature and be able to code species accordingly. You simply refer to "social grouping between adults and juveniles". What do you mean by association: do individuals have to be found within one body distance, in the same home range, within the same cave, within the same general area? How long do associations have to last: if it were to take females time to give birth to large numbers of live offspring such that early-borns are already foraging while females are still giving birth, does this count as an association? How are associations different from extended parental care? What about associations that only involve adults, such as congregations for mating or hibernation?

Similarly, what does egg attendance mean? What about instances where females defend a territory, lay their eggs within that territory, and use the same area as where the eggs are deposited? How long does egg attendance have to last? As above, would instances where some offspring hatch early while mothers still attend unhatched eggs count as a social grouping under your definition?

In addition, the second sentence of your abstract highlights the high 'functional diversity' of sociality in this group, but in the following you only provide one definition and lump all species together. What is this functional diversity? Is it justified to combine all types of social grouping in one analyses, that is, do they most likely reflect homologous or analogous expressions? I think you need to add more details, both in the main text and in the

methods, to both help the reader understand sociality in this clade and indicate the validity of your classification.

Approaches:

The manuscript includes a number of diverse, state-of-the art approaches to investigate the evolutionary history of social grouping and its relationships with live-bearing and egg attendance. However, this means that detail is lost in explaining why a particular approach is relevant for answering a particular question and how exactly analyses were performed. The results of all the different models are currently presented in a single paragraph, making it difficult to follow why these approaches are appropriate to address your predictions, and why you used different approaches. You need to expand this section to allow a broad readership to understand what you did.

In line 61 you state that there is “considerable phylogenetic structure in the distribution of social grouping”. Why does this matter? What does it mean? Do you have some kind of estimate how often social grouping might have evolved? Are there large clades of social species?

In line 66 you mention differential rates of speciation and extinction. Why does this matter? Does live-bearing in fact influence speciation rates?

In line 68 you state that “ancestral state reconstruction supported the correlated evolution of social grouping and viviparity”. For someone who is not familiar with phylogenetic approaches, how did the ancestral state reconstruction support this hypothesis?

Figure 3 is somewhat meaningless – it compares two single numbers. It also lacks context: to really determine whether there is a significant association between viviparity and transitions to group living we need to know how often these transitions could have occurred by chance. If 80% of all species are viviparous, the observation that there are four times as many transitions to group living in viviparous species than in oviparous ones simply reflect chance.

Causality:

Given that there is no obvious known mechanisms that links live-bearing to social grouping I think you need to discuss alternatives that do not assume a direct causal link between the two traits. For example, seasonal breeding might influence longevity such that females are more likely to give birth to live offspring and lead to extended juvenile periods and reduced competition as mothers are not investing into reproduction while previous offspring are present. Alternatively, territoriality might be associated with stable food supplies that allow females to produce live-offspring, and as you state, mothers defending territories and remaining on them for multiple breeding seasons will lead to a localization of offspring within parental territories. I think in this context it would be helpful to have more information on the ‘functional diversity’ and on some of the exceptions, where social groupings appear to have evolved in non-viviparous species. These might be interesting

candidates to understand potential pathways linking these traits.

As mentioned above, I think the authors should have the material to address all these comments. A revised manuscript is likely to be of interest to a wide readership.

Reviewer #1 (Remarks to the Author):

Evolution of sociality is subject of much interest and reptiles so far were quite poorly represented in phylogenetic analyses of sociality. A major strength of this paper that it takes up this challenge and using impressive dataset and methodology, proposes a link between live bearing and sociality.

The topic is of general interest, the analyses are professionally done and the writing-up is excellent.

I have 4 comments.

1a. Data extraction was focused on traits related to parental care eg brood defence, provisioning and parental care. These traits are only a small subset of "sociality", so I'm confused why the focus of the paper is on sociality rather than on care.

Response: The reviewer is correct that parental care is central to our analyses. However, as stated in the paper (lines 31 - 34; 37- 40), the implications in this case are much broader. Fraternal social groups, in which social groupings emerge as a result of related individuals staying together to form groups, is underpinned in most systems initially by an increase in parent-offspring associations (e.g., refs.^{1,2}). For example, it is currently thought that cooperative societies evolve from bi-parental units where offspring stay to help raise siblings, with whom they share genes. Therefore, if we are to understand the origins of sociality *per se* we must first ask questions relating to when and how social interactions result in stable parental bonds and social groups. Our paper does exactly this, first by investigating the extent to which parent-offspring associations are promoted by particular types of life history traits (i.e., viviparity and egg attendance) and secondly, by then asking if those same life history traits also facilitated the emergence of stable social units, similar to those seen in birds and mammals (lines 146 - 160). Thus, although parental care is a key trait, the paper itself and specifically components of the analysis expand well beyond parental care and towards a broader focus on understanding the origins of family living/sociality itself. We have now clarified this in the introduction (lines 37 – 64) the discussion (lines 195 - 206) and the methods (lines 239 – 247).

1b. Social grouping just confuses this issue, and it is never made clear what's the difference between sociality and "stable social grouping".

Response: We thank the reviewer for highlighting this oversight – we now define stable social grouping on lines 147-149 of the manuscript and provide additional explanation and references in the methods section (lines 281 - 285).

1c. For a non-herpetologist all of these traits represent various aspects of caring, and there is a bit of leap in logic to sociality.

Response: Please see response to point 1a above. Stable social organisation in lizards emerges from a prolonged association between males and females (long-term stable pair bonds) and from a prolonged association between parents and offspring. Thus, fundamental to understanding the evolution of sociality in this system, and in general, are the factors which may facilitate these prolonged associations. We test the

prediction here that viviparity is one of those factors. Indeed, viviparity should increase the level of association between parents and offspring, thus creating novel selective environments from which more complex forms of social behaviour and social grouping emerge. Our data suggests this is the case. We have now clarified these points in the introduction (Lines 31 – 40) and discussion (Lines 169 – 171; 195 – 206).

1d. Reptiles could be considered having some sort of sociality, although this seems strikingly different from the sociality seen in fishes, birds and mammals.

Response: We disagree with the reviewer on this point. Sociality in reptiles is actually strikingly similar to that observed in fish, birds and mammals, albeit simpler in some respects. Indeed, a large number of reptile species across a range of reptile groups form stable social aggregations based around kin (see reviews in refs.³⁻⁶). These family groups are very similar, in some cases identical, to those observed in birds, mammals, and fish. Indeed, it is likely that convergent processes have resulted in the emergence of family living in reptiles. Specifically, complex social organisation emerges when ecological conditions and life history characteristics act together to impose constraints that make close kin interact. What sets reptiles apart is the opportunity they provide to test these predictions (lines 42 - 64). This is exactly what we test here and what we believe makes this paper so novel and powerful. We have amended the introduction to make this point clearer (Lines 53 - 64).

2. Numerous species that were scored "social" seem to exhibit a simple trait: egg-manipulation. It is a bit fuzzy what this trait means and even less clear whether this behaviour would classify parental care at all.

Response: Please note that, while some species coded as displaying social grouping *also* displayed egg attendance behaviour, this was not the justification for their being coded as displaying social grouping: all species coded as displaying social grouping were done so based on specific reports of aggregations of adults and juveniles. We apologise for this confusion and have amended the MS to both provide a clearer description of egg attendance behaviour (Supp Info: Table 4, ‘Quantifying the functional and taxonomic diversity of care’ section) and clarify our criteria for coding species as displaying social grouping (Revised MS: lines 272 – 276; Supp Info: ‘Social Grouping’ section).

3. Reporting bias is an issue in understudied groups like amphibians and reptiles. I wonder whether the cut-off line (>100 publications) is the best way of filtering out pseudo-absences in traits. For example, a species can be very well studied from taxonomic point of view although there may be still no detailed study on its breeding system so that the traits of interest could be easily missed.

Response: The reviewer points toward a general challenge in phylogenetic comparative analyses, with data absences and incomplete reporting representing an almost ubiquitous feature of such studies^{7,8}. Thus, researchers are at times forced to make decisions regarding the coding of species included in analyses. Our threshold of 100 citations is in fact more conservative than recent comparable studies (e.g., ref.⁹: West and Capellini 2016, Nat Commun; ref.¹⁰: Dey et al. 2017, Nat Ecol Evol). More importantly, our study incorporates information on the reproductive biology of

species whenever available to inform decisions regarding the coding of absence of social grouping, and our use of multiple exclusion criteria (Supp Info: 'Assigning absence of social grouping' section) provides supporting evidence for most species, as only 23 species were assigned absence of social grouping based solely on having >100 citations.

4. Science is moving toward data sharing and open access, and I find it weird that the authors do not intend making their data publicly available. This is not acceptable and would prevent others from checking the data and potentially following up the study. The dataset and their sources should be made available in open access data depository like Dryad.

Response: We are not sure how we gave the reviewer the impression that we would not be making our data publically available upon acceptance, as this is certainly our intention. We would also be more than happy to share our data with the editor and the reviewer prior to acceptance if the editor deems this necessary.

Reviewer #2 (Remarks to the Author):

In this paper, the authors assess the evolution of sociality across squamate reptile. In addition to providing a further, detailed description of variability in sociality across this thus far understudied group of animals, their results indicate that giving birth to live young strongly increases the chance that groups with overlapping generations form. This manuscript adds to a growing literature of comparative studies that indicate that sociality tends to be rare, and generates insight that are not only relevant for this particular clade but for the evolution of sociality in animals more generally.

My problem with the current version of the manuscript is that, for me, this very short manuscript leaves out crucial information both for people not familiar with the subject and those potentially wanting to build on this study. Adding further detail will help readers to understand definitions and how they compare to other taxonomic groups, be able to replicate their approaches, and clarify potential causality underlying the detected links. This is an impressive data-set (though I have not checked the actual classification of species as I am not familiar with this group of animals), the analyses appear solid, and the authors are careful in not drawing speculative conclusions. My comments are mainly to provide additional clarifications and information that might help readers to better follow the arguments.

Response: We thank the reviewer for these comments. Our initial aim was to write this paper as a brief communication and as a result we perhaps left out some context that would benefit readers not familiar with the study area. We have now rectified this in the revised version of the MS and believe that this makes the paper a much stronger piece of work (lines 37 - 64; also see revised methods section)

Definitions:

Neither the main text nor the method contain a sufficient definition of "sociality" that would allow me to go to the primary literature and be able to code species accordingly. You simply refer to "social grouping between adults and juveniles". What do you mean by association: do individuals have to be found within one body

distance, in the same home range, within the same cave, within the same general area?

Response: We thank the reviewer for raising this issue. We have amended the manuscript (lines 258 – 279) to contain a more thorough description of our inclusion criteria and also point the reader toward details provided in the supplementary information (Supp Info: Table 2, Table 4, throughout ‘Experimental Procedures’ section). As it is so rarely reported, we did not impose a specific distance between adults and juveniles necessary to demonstrate social grouping, but instead relied on author’s interpretations of grouping behaviour, e.g., species with descriptions of adults and juveniles forming long- or short-term associations (based on delayed post-natal dispersal), sharing crevice or refuge sites, seen ‘basking together’, or described as forming ‘aggregations’ or ‘groups’, were considered to display social grouping. These methods are standard in comparative research and consistent with recent comparable studies (e.g., ref.⁹: West and Capellini 2016, Nat Commun), but have now been clarified in the Experimental Procedures section of the revised Supp Info document.

How long do associations have to last: if it were to take females time to give birth to large numbers of live offspring such that early-borns are already foraging while females are still giving birth, does this count as an association? How are associations different from extended parental care?

Response: The length of parent-offspring associations can vary considerably, from several days, to months or even years^{4,6}. We have not included any data that result from birth itself being prolonged (and almost all lizards give birth to all young at once). For example, species in which offspring emerge in the presence of parents but disperse within hours were not included as displaying social grouping. However, data on the specific length of associations is available for only the most thoroughly studied species and we therefore we did not require that the specific duration of parent-offspring associations be reported to code a species as displaying social grouping. In many cases, these social associations are likely to represent a form of extended parental care, although experimental confirmation is rare (but see refs.¹¹⁻¹³).

What about associations that only involve adults, such as congregations for mating or hibernation?

Response: The reviewer is correct in pointing out that several squamate species form aggregations around particular resources such as hibernacula or landscape features with desirable thermal characteristics. These aggregations are primarily egalitarian, transient and do not result in the emergence of strong bonds between individuals (See refs.^{5,6} for review). Such aggregations were not the focus of this study (lines 263 – 264). Instead we were interested in social aggregations based on parents and offspring remaining together, in line with the extensive comparative work carried out on family living species in other taxa (e.g., ref.¹⁴).

Similarly, what does egg attendance mean?

Response: We define egg attendance as any behaviour that involves parental contact or manipulation of eggs (Supplementary Table 4 of submitted manuscript). We have

now amended this description to contain greater detail (Supplementary Table 4 of revised manuscript).

What about instances where females defend a territory, lay their eggs within that territory, and use the same area as where the eggs are deposited? How long does egg attendance have to last? As above, would instances where some offspring hatch early while mothers still attend unhatched eggs count as a social grouping under your definition?

Response: The situation described by the reviewer would not count as egg attendance (Supp Table 4). As described above, social groupings, in terms of prolonged parent-offspring associations require parents and offspring to associate with one another for longer than the birthing/hatching period. This has now been clarified in the main body of the text (lines 272 - 276 of the submitted manuscript).

In addition, the second sentence of your abstract highlights the high 'functional diversity' of sociality in this group, but in the following you only provide one definition and lump all species together. What is this functional diversity?

Response: We briefly describe how social groups vary in size, form and duration as well as whether juveniles associate with adults of one or both sexes in the revised manuscript (lines 87 – 98). However, we have now removed this reference to 'functional diversity' in the abstract, as this is not the focus of the current paper, and is addressed more thoroughly in previous publications (i.e., refs.^{4,6}).

Is it justified to combine all types of social grouping in one analyses, that is, do they most likely reflect homologous or analogous expressions?

Response: We believe it is. However, the extent to which social grouping would count as a homology or an analogy is not obvious; the same issue arises for, e.g., cooperative breeding in birds. We do not take a stand on this issue. What is important in the present context is that, once they arise, social associations between parents and offspring can set the stage for the evolution of more complex forms of parental behaviour and also social organisation (see Introduction, lines 31 - 34; 37 - 40, Discussion, lines 195 - 206).

I think you need to add more details, both in the main text and in the methods, to both help the reader understand sociality in this clade and indicate the validity of your classification.

Response: As well as the amendments outlined above, we have now added additional detail throughout the MS to justify the validity of our classification (lines 55 - 61; 84 - 95; 147 - 149).

Approaches:

The manuscript includes a number of diverse, state-of-the art approaches to investigate the evolutionary history of social grouping and its relationships with live-bearing and egg attendance. However, this means that detail is lost in explaining why a particular approach is relevant for answering a particular question and how

exactly analyses were performed. The results of all the different models are currently presented in a single paragraph, making it difficult to follow why these approaches are appropriate to address your predictions, and why you used different approaches. You need to expand this section to allow a broad readership to understand what you did.

Response: Thank you, this is very helpful. We now provide additional detail within both the results (lines 116 – 120) and methods sections (lines 293 - 395) regarding the justification for each of our methodological approaches and how they allow us to comprehensively address our research question.

In line 61 you state that there is “considerable phylogenetic structure in the distribution of social grouping”. Why does this matter? What does it mean? Do you have some kind of estimate how often social grouping might have evolved? Are there large clades of social species?

Response: This statement refers to the significant s^2 value returned by binary phylogenetic generalized linear mixed modelling of the relationship between parity mode and social grouping, performed in the R package ‘ape’. For example, $s^2 = 0$ implies no phylogenetic signal and therefore that the distribution of the trait is random with respect to tree topology (i.e. is not influenced by the phylogenetic relationships between species). As the value of s^2 increases this implies that an increasing proportion of variance in the distribution of the trait is explained by tree topology. This is now clarified on lines 300 - 306.

Our analyses revealed several clades that stand out as containing many social species, particularly the Australian Egeriinae and North American Crotalinae (lines 92-98). As indicated by significant s^2 values, the relatively common occurrence of social grouping within these clades most likely reflects divergence and speciation from a shared ancestor that displayed social grouping, rather than multiple independent origins of the trait.

In line 66 you mention differential rates of speciation and extinction. Why does this matter? Does live-bearing in fact influence speciation rates?

Response: The reviewer is correct, it has been suggested that live bearing may have influenced rates of speciation within the squamates¹⁵. Thus, we considered this a potentially important source of variation in the phylogenetic distribution of social grouping and concluded that best practice would be to also perform our hypothesis test using a modelling framework capable of accounting for state specific rates of speciation and extinction. This is now explicitly stated in the results (lines 116 - 119) and methods sections (lines 335 - 339).

In line 68 you state that “ancestral state reconstruction supported the correlated evolution of social grouping and viviparity”. For someone who is not familiar with phylogenetic approaches, how did the ancestral state reconstruction support this hypothesis?

Response: Our ancestral state reconstructions support correlated evolution between viviparity and social grouping because reconstructed histories of the two traits indicate much greater similarity, in terms of the timing and phylogenetic position of

transitions, than expected by chance (lines 345 - 347, Figure 2B). Furthermore, visualising these reconstructions tentatively suggests a causal directionality between the two traits: the posterior probability density of edge states across all 1000 character maps for each trait suggest that the evolution of viviparity preceded the evolution of social grouping in most independent origins of the trait (Figure 2A). Specifically, red edges in the left hand tree (points in the evolutionary history of clades at which there is high confidence that viviparity had already evolved) are more deeply nested, and therefore occurred earlier, than those in the right hand tree (points at which there is high confidence that social grouping had evolved), suggesting that the presence of viviparity may have promoted the emergence of social grouping across multiple independent events. We have clarified this in the revised MS on lines 120 – 123.

Figure 3 is somewhat meaningless – it compares two single numbers. It also lacks context: to really determine whether there is a significant association between viviparity and transitions to group living we need to know how often these transitions could have occurred by chance. If 80% of all species are viviparous, the observation that there are four times as many transitions to group living in viviparous species than in oviparous ones simply reflect chance.

Response: We believe the reviewer may have misinterpreted this result: 80% of squamates are in fact *oviparous* (line 109), therefore the observation that, proportionately, there have been many times more transitions to social grouping in viviparous clades is very unlikely under the null expectation of no difference in transition rates between parity modes. This suggests a strong association between live bearing and social grouping. Figure 3 however deals with transitions to stable social grouping. The points plotted on this graph represent the average number of transitions to stable social grouping from each parity mode based on 1000 permuted ancestral reconstructions and therefore provide good evidence that transitions to stable forms of social grouping in squamates are also associated with viviparity. Indeed, this is a key result of our research: not only does viviparity appear to precede the evolution of social grouping, it also appears to facilitate the transition to more stable, kin-based social grouping and therefore the emergence of family life (similar to that observed in other taxa). Having said that, we have now re-structured the results in line with previous comments by the reviewers and agree with the reviewer that the data represented in the figure could be better incorporated into the main body of text. Thus, we now simply report the number of transitions estimated from these analyses in the text (line 159) and have moved the graph to the supplementary information (Supp Info: Figure 3).

Causality:

Given that you there is no obvious known mechanisms that links live-bearing to social grouping I think you need to discuss alternatives that do not assume a direct causal link between the two traits. For example, seasonal breeding might influence longevity such that females are more likely to give birth to live offspring and lead to extended juvenile periods and reduced competition as mothers are not investing into reproduction while previous offspring are present. Alternatively, territoriality might be associated with stable food supplies that allow females to produce live-offspring, and as you state, mothers defending territories and remaining on them for multiple breeding seasons will lead to a localization of offspring within parental territories. I think in this context it would be helpful to have more information on the 'functional

diversity' and on some of the exceptions, where social groupings appear to have evolved in non-viviparous species. These might be interesting candidates to understand potential pathways linking these traits.

Response: We clearly outlined several key causal mechanisms that link live-bearing to social groupings (lines 169 – 194 in the revised version of the MS). In fact the most parsimonious causal mechanism is that live birth simply facilitates a greater level of association between parents and offspring than does egg laying (which is the key premise of the study, highlighted in the introduction; lines 31 - 34). The reviewer is completely correct, however, that while traits such as viviparity may pre-dispose species to exhibit parent-offspring association, it is ecology that ultimately dictates whether this probability is realised. Furthermore, we agree that additional life history traits, such as longevity and delayed maturity, may also be important, as they will influence the cost-benefits of offspring remaining within the parental home range (as has been argued for birds and mammals; refs.^{14,16,17}). We have now expanded the discussion of our results to provide some additional explanations for what ultimately allows the possibility of greater parent-offspring associations offered by viviparity to be realised (lines 207 – 229). However, this does not take away from the fact that viviparity will be, as we show conclusively, fundamentally important to this process.

As mentioned above, I think the authors should have the material to address all these comments. A revised manuscript is likely to be of interest to a wide readership.

Response: We thank the reviewer for his/her constructive comments and hope that our revisions have added the extra context required.

References

1. Queller, D. C. Extended Parental Care and the Origin of Eusociality. *Proc. Biol. Sci.* **256**, 105–111 (1994).
2. Field, J. & Brace, S. Pre-social benefits of extended parental care. *Nature* **428**, 650–652 (2004).
3. Doody, J. S., Burghardt, G. M. & Dinets, V. Breaking the Social-Non-social Dichotomy: A Role for Reptiles in Vertebrate Social Behavior Research? *Ethology* **119**, 95–103 (2013).
4. While, G. M., Halliwell, B. & Uller, T. in *Reproductive Biology and Phylogeny of Lizards and Tuatara* 590–619 (CRC Press, 2015).
5. Gardner, M. G., Pearson, S. K., Johnston, G. R. & Schwarz, M. P. Group living in squamate reptiles: a review of evidence for stable aggregations. *Biol Rev Camb Philos Soc* **91**, 925–936 (2016).
6. Whiting, M. & While, G. M. in *Comparative Social Evolution* (eds. Rubenstein, D. R. & Abbot, P.) 390–426 (2017).
7. Freckleton, R. P. The seven deadly sins of comparative analysis. *J. Evol. Biol.* **22**, 1367–1375 (2009).
8. Nakagawa, S. & Freckleton, R. P. Missing inaction: the dangers of ignoring missing data. *Trends Ecol. Evol. (Amst.)* **23**, 592–596 (2008).
9. West, H. E. R. & Capellini, I. Male care and life history traits in mammals. *Nature Communications* **7**, 1–10 (2016).
10. Dey, C. J. *et al.* Direct benefits and evolutionary transitions to complex societies. *Nat. ecol. evol.* **1**, 0137–8 (2017).
11. O'Connor, D. E. & Shine, R. Parental care protects against infanticide in the lizard *Egernia saxatilis* (Scincidae). *Animal Behaviour* **68**, 1361–1369 (2004).
12. Hoss, S. K., Deutschman, D. H., Booth, W. & Clark, R. W. Post-birth separation affects the affiliative behaviour of kin in a pitviper with maternal attendance. *Biological Journal of the Linnean Society* **116**, 637–648 (2015).
13. Botterill-James, T. *et al.* Habitat structure influences parent-offspring association in a social lizard: implications for understanding the origins of parental care. *Frontiers in Ecology and Evolution*
14. Covas, R. & Griesser, M. Life history and the evolution of family living in birds. *Proc. Biol. Sci.* **274**, 1349–1357 (2007).
15. Pyron, R. A. & Burbrink, F. T. Early origin of viviparity and multiple reversions to oviparity in squamate reptiles. *Ecol Lett* **17**, 13–21 (2013).
16. Koenig, W. D., Pitelka, F. A., Carmen, W. J., Mumme, R. L. & Stanback, M. T. The Evolution of Delayed Dispersal in Cooperative Breeders. *The Quarterly Review of Biology* **67**, 111–150 (1992).
17. Clutton-Brock, T. H. *The Evolution of Parental Care*. (Princeton University Press, 1991).

Reviewers' comments:

Reviewer #1 (Remarks to the Author):

This is a well-rounded revision. The rationale and theoretical framework underpinning the comparative analyses are well explained, and various ambiguities have been clarified. I concur with the authors that this work is timely and nicely follow up a recent review on the topic (Gardner et al. 2016). I accept the arguments in regards to parental care vs social evolution.

I have 3 reservations.

1. Fig 2 appears to show large number of species that have no social information available. According to the Appendix 1223 species have social grouping data available, and given that the objective of the paper is to test sociality vs parity, I'm confused why Fig 2 shows lot more species?

2. Given the often quite circumstantial evidence of "stable" social grouping eg snapshot records of young and adults of occasionally both sexes in the same location, the implication of the work needs to be toned down. Some of these casual reports that seem to give rise to the use of "stable" grouping are far cry from the well-studied mammalian and avian systems where researchers monitor clans, troops or groups of marked individuals throughout a long time period, often over many years.

Therefore, statements such as on lines 166-167 should be toned down substantially eg "...many of which represent kin-based sociality POSSIBLY comparable to that seen in mammals and birds."

3. Table 1 is not really user-friendly. One problem is that it holds too much information: the key results as well as the results of the sensitivity analyses. There must be smarter ways of splitting the two so that keeping the key results in the main paper, and relegating the sensitivity analyses to supplementary material.

Reviewer #2 (Remarks to the Author):

The authors have done a great job revising the paper. The added explanations and definitions make the focus and any potential limitations clearer: for example, while sociality might be defined it is clear what the term means here and that it has biological relevance for the question. The method and result section now also appear clearer to follow and understand.

I have only one minor remaining comment: in the second paragraph of the introduction, the authors argue that birds and mammals, the taxonomic groups that have usually been the focus for the evolution of sociality, are not well suited to address the specific hypothesis

about a link between live-bearing and extended associations between parents and offspring. However, the reason is slightly awkwardly formulated: stating that parent-offspring associations are ubiquitous in both lineages even though all birds lay eggs and almost all mammals are viviparous raises the question whether the hypothesis is valid. I think this needs rewording, and it might be helpful to return to the broader taxonomic focus in the discussion where you mention that egg attendance is difficult to measure and extended egg attendance might be associated with parent-offspring association in non-viviparous lizards. My comment would only require a limited rewording of the introduction and the discussion, and I think this manuscript is ready to be published.

Reviewer #3 (Remarks to the Author):

This is a very large phylogenetic comparative study with multiple analyses of more than 1000 species of squamate reptiles that strongly concludes that wherein social groups involving adults and juveniles have evolved multiple times, viviparity, but not egg attendance, appears to have facilitated the emergence of social associations. The results are very convincing and this study greatly adds to our understanding of the evolution of complex sociality. I think it is an important contribution to the field of the evolution of sociality. However, I don't think the authors are the first to reason that viviparity may lead to conditions favoring the evolution of parent-offspring social behavior (I've always thought there was an association). They should cite and describe previous thinking along these lines. Even so, theirs is a much bigger and much more formal test than past studies of viviparity as a precursor to the evolution of sociality in reptiles.

L 41-51: It is obviously true that the investigation of precursors like viviparity to complex sociality cannot well be studied in birds or mammals since all birds lay eggs and most (all?) species care for their young, and all mammals are viviparous and all care for their young. This evolutionary transition to complex sociality needs to be addressed in some other taxon that shows greater trait variation and a good one is squamate reptiles, as the authors point out. However, I don't believe they are the first authors to make this point (e.g., see Shah et al. 2003) and they need to dig deeper into the literature to cite authors before them making this same excellent point.

L 116-122: Very good that authors considered that the evolution of viviparity can accelerate speciation rates and thus alter the tree topology itself and subsequently appear to be a more direct explanatory effect than it really is. Nevertheless, their analyses accounting for this effect still showed correlated evolution of social grouping and live bearing.

Methods: I was impressed with the two-layered search protocol to identify parental care behavior and social grouping, and also searching for presence of stable social grouping, which is an even more advanced form of simple sociality. I think the analyses of two datasets, conservative and relaxed, as described in lines 380-394, is also a good idea and using absence at the family level of sociality so as to be able to include more species (with social grouping = 0) in the analysis is reasonable, especially since the same analyses were

also performed on the conservative dataset.

Supplementary material: I think the authors applied reasonable criteria to assign the absence of social grouping to a species. They assigned SG as being absent in a species if it met one of the following three criteria: 1) Behavioral forms of parental care other than Parent-Offspring Association had been reported with no mention of associations between adults and juveniles in any of the literature accessed. 2) Studies of life history, reproductive ecology, spatial ecology, or habitat use were available in which researchers conducted observations and/or field collections (not using passive capture techniques) during periods of hatching or parturition, collected observations of both adults and juveniles, but did not report any social association between these age classes. 3) The species is considered to be well studied, defined as having a citation count of ≥ 100 peer reviewed publications, making it highly unlikely that such behavior would not have been reported if present. However, the third criterion is a little sketchy. If most of these ≥ 100 studies including a species are taxonomic ones or especially just inclusion in some phylogenetic study, then one can't really conclude that the species behavioral ecology is well studied. Can the authors speak to this possible weakness? --I see that a previous reviewer also made this same point and I think at least a part of the authors' rebuttal should be included in the manuscript.

Finally, because I am a behavioral ecologist studying social behavior in lizards, I am not strong in the phylogenetic comparative analyses described in the manuscript. I asked and received permission from the journal editor to consult with a departmental colleague, who in fact does these kinds of analyses in his own research. This is what he had to say:

(1) I am not at all familiar with the PGLMMs. I could, however, evaluate the MuSSE and ancestral-state reconstructions. The latter two seemed well done. That said, the authors describe (in the main methods) the PGLMMs as though it were a standard procedure, giving only ONE citation for the whole thing. That gives the impression that PGLMMs are standard, but I've honestly not read one single methods paper that introduces them, uses them, or tests their properties. So the authors need to better justify their use of this method and give readers citations to follow for their own use of the methods.

(2) They only used one phylogeny. Why just this one? Are the results robust? Also, they say it was the Pyron et al. 2013 phylogeny, but their figures show that they used the time-calibrated phylogeny of Pyron and Burbrink 2014. The two trees have the same topology, but the former has branch lengths in units of genetic substitution. Despite this, the authors cite the former as their "time-calibrated phylogeny" on lines 348-349. So they should cite Pyron and Burbrink 2014 (the real time-calibrated tree) as well.

(3) MuSSE methods and Supp. table 3: In the methods the authors say that they tested their free-parameter model (in which all parameters were free to vary) with "a constrained version of the model in which these transition rates were forced to be equal". This description is a little misleading, because the only model they test in all these cases is whether a transition TO social grouping is higher in viviparous than oviparous lineages. Is it reasonable to also ask about the LOSS of social grouping? I think it's reasonable to expect that if viviparity makes it more likely to become social, then oviparity should make it more

likely to lose sociality. They don't test this latter possibility (which would be testing if q21 is higher than q43).

The flip side of this question is to ask about this correlation. That is, could being social lead to viviparity? Maybe that doesn't make mechanistic sense, but again, if it is possible, they do not test that possibility (i.e. test whether q23 is higher than q14). I would ask them to be more explicit about what they actually did and to justify why they didn't test these other possibilities (i.e. a biological justification for why testing them does not make sense).

Shah, B., R. Shine, S. Hudson, and M. Kearney. 2003. Sociality in lizards: why do thick-tailed geckos (*Nephrolepis*) aggregate? *Behaviour* 140, 1039-1052.

Response to Reviewers' comments

Reviewer #1 (Remarks to the Author):

This is a well-rounded revision. The rationale and theoretical framework underpinning the comparative analyses are well explained, and various ambiguities have been clarified. I concur with the authors that this work is timely and nicely follow up a recent review on the topic (Gardner et al. 2016). I accept the arguments in regards to parental care vs social evolution.

I have 3 reservations.

1. Fig 2 appears to show large number of species that have no social information available. According to the Appendix 1223 species have social grouping data available, and given that the objective of the paper is to test sociality vs parity, I'm confused why Fig 2 shows lot more species?

Response: The reviewer is correct; this figure displays the entire phylogeny of Pyron et al. 2014 (3951 spp.). Originally we thought it would be informative to plot the distribution of current data across the entire tree, but have reconsidered this approach and now provide a revised figure including only the 1210 species included in analyses. Please note that although social grouping data were available for 1224 species, 14 species were excluded from analyses because they were not present in the phylogeny or were reported as displaying both viviparity and oviparity, explaining this discrepancy in species number (See Supplementary Data File 1).

2. Given the often quite circumstantial evidence of "stable" social grouping eg snapshot records of young and adults of occasionally both sexes in the same location, the implication of the work needs to be toned down. Some of these casual reports that seem to give rise to the use of "stable" grouping are far cry from the well-studied mammalian and avian systems where researchers monitor clans, troops or groups of marked individuals throughout a long time period, often over many years.

Therefore, statements such as on lines 166-167 should be toned down substantially eg "...many of which represent kin-based sociality POSSIBLY comparable to that seen in mammals and birds."

Response: We acknowledge the reviewers reservations and have clarified the comparison between the stable social grouping observed in reptiles and that observed in birds and mammals, lines 71-73 and 170-172 of the revised manuscript. Nevertheless, we believe that a comparison between the stable social grouping seen in squamates and other vertebrate groups is both valid and appropriate. As outlined on lines 280-285 of the previous manuscript, our criteria for classifying stable social grouping (which is what the lines outlined above relate to) follows that set out by Gardner et al. 2016, conservatively defining stable social groups as either:

- 1) Permanent aggregations, i.e., social groups that are maintained consistently throughout the year, or;
- 2) Where aggregations are periodic or seasonal, social groups in which individuals maintain group membership across multiple seasons or years.

To be fulfilled, these criteria require observations from prolonged and sustained fieldwork that demonstrate stability or consistent membership of social groups (including pair bonds) through time. Such fieldwork goes far beyond chance observations of young and adults seen in the same location (e.g., ref. ¹⁻⁵). Indeed, much of the work on stable social aggregations in lizards is based on exactly the kind of long-term data that is available for other vertebrates that the reviewer points out. For example, Mike Bull's work on the pair living *Tiliqua rugosa* has been tracking individuals for the past 30+ years and our own work on the family living *Liopholis whitii* has involved yearly sampling of the same population for 14 years.

3. Table 1 is not really user-friendly. One problem is that it holds too much information: the key results as well as the results of the sensitivity analyses. There must be smarter ways of splitting the two so that keeping the key results in the main paper, and relegating the sensitivity analyses to supplementary material.

Response: We have now amended Table 1 to contain only model estimates and associated test statistics and provide a fully expanded version of the table including sensitivity analyses in the supp info (Supplementary Table 4).

Reviewer #2 (Remarks to the Author):

The authors have done a great job revising the paper. The added explanations and definitions make the focus and any potential limitations clearer: for example, while sociality might be defined it is clear what the term means here and that it has biological relevance for the question. The method and result section now also appear clearer to follow and understand.

I have only one minor remaining comment: in the second paragraph of the introduction, the authors argue that birds and mammals, the taxonomic groups that have usually been the focus for the evolution of sociality, are not well suited to address the specific hypothesis about a link between live-bearing and extended associations between parents and offspring. However, the reason is slightly awkwardly formulated: stating that parent-offspring associations are ubiquitous in both lineages even though all birds lay eggs and almost all mammals are viviparous raises the question whether the hypothesis is valid. I think this needs rewording, and it might be helpful to return to the broader taxonomic focus in the discussion where you mention that egg attendance is difficult to measure and extended egg attendance might be associated with parent-offspring association in non-viviparous lizards. My comment would only require a limited rewording of the introduction and the discussion, and I think this manuscript is ready to be published.

Response: We have now amended the introduction (lines 41-52) to make this point more explicit.

Reviewer #3 (Remarks to the Author):

This is a very large phylogenetic comparative study with multiple analyses of more than 1000 species of squamate reptiles that strongly concludes that wherein social groups involving adults and juveniles have evolved multiple times, viviparity, but not egg attendance, appears to have facilitated the emergence of social associations. The results are very convincing and this study greatly adds to our understanding of the evolution of complex sociality. I think it is an important contribution to the field of the evolution of sociality. However, I don't think the authors are the first to reason that viviparity may lead to conditions favoring the evolution of parent-offspring social behavior (I've always thought there was an association). They should cite and describe previous thinking along these lines. Even so, theirs is a much bigger and much more formal test than past studies of viviparity as a precursor to the evolution of sociality in reptiles.

Response: We thank the reviewer for their valuable comments. The reviewer is quite right that we are not the first authors to draw a connection between viviparity and natal conditions that may promote the emergence of kin-based social interactions (e.g. ref. ^{2,6,7}). Indeed, we discuss this in reference to relevant literature on lines 36-39, 168-172, 195-198 and 224-227 of the previous manuscript. It was not our intention to suggest that this was an idea purely of our own conception and have now added additional references to the statement on lines 36-39 of the revised manuscript to make the precedence for this test clearer.

L 41-51: It is obviously true that the investigation of precursors like viviparity to complex sociality cannot well be studied in birds or mammals since all birds lay eggs and most (all?) species care for their young, and all mammals are viviparous and all care for their young. This evolutionary transition to complex sociality needs to be addressed in some other taxon that shows greater trait variation and a good one is squamate reptiles, as the authors point out. However, I don't believe they are the first authors to make this point (e.g., see Shah et al. 2003) and they need to dig deeper into the literature to cite authors before them making this same excellent point.

Response: The reviewer is quite right that we are not the first authors to identify the limitations of using mammals and birds to test hypothesis about the evolutionary origins of sociality ^{2,8} or even that squamate reptiles provide an excellent taxa in this regard (e.g. ref. ^{2,6-8}). We have added some of these references to our statement on lines 43-45. The point we make on lines 45-52 speaks specifically to our analysis of the links between parity mode, egg attendance and social grouping, so we have amended this section to make our particular supposition more explicit.

L 116-122: Very good that authors considered that the evolution of viviparity can accelerate speciation rates and thus alter the tree topology itself and subsequently

appear to be a more direct explanatory effect than it really is. Nevertheless, their analyses accounting for this effect still showed correlated evolution of social grouping and live bearing.

Response: We thank the reviewer for recognising the hard work that went in to these analyses!

Methods: I was impressed with the two-layered search protocol to identify parental care behavior and social grouping, and also searching for presence of stable social grouping, which is an even more advanced form of simple sociality. I think the analyses of two datasets, conservative and relaxed, as described in lines 380-394, is also a good idea and using absence at the family level of sociality so as to be able to include more species (with social grouping = 0) in the analysis is reasonable, especially since the same analyses were also performed on the conservative dataset.

Supplementary material: I think the authors applied reasonable criteria to assign the absence of social grouping to a species. They assigned SG as being absent in a species if it met one of the following three criteria: 1) Behavioral forms of parental care other than Parent-Offspring Association had been reported with no mention of associations between adults and juveniles in any of the literature accessed. 2) Studies of life history, reproductive ecology, spatial ecology, or habitat use were available in which researchers conducted observations and/or field collections (not using passive capture techniques) during periods of hatching or parturition, collected observations of both adults and juveniles, but did not report any social association between these age classes. 3) The species is considered to be well studied, defined as having a citation count of ≥ 100 peer reviewed publications, making it highly unlikely that such behavior would not have been reported if present. However, the third criterion is a little sketchy. If most of these ≥ 100 studies including a species are taxonomic ones or especially just inclusion in some phylogenetic study, then one can't really conclude that the species behavioral ecology is well studied. Can the authors speak to this possible weakness? --I see that a previous reviewer also made this same point and I think at least a part of the authors' rebuttal should be included in the manuscript.

Response: This is the relevant response to the previous reviewer's comment:

“The reviewer points toward a general challenge in phylogenetic comparative analyses, with data absences and incomplete reporting representing an almost ubiquitous feature of such studies^{9,10}. Thus, researchers are at times forced to make decisions regarding the coding of species included in analyses. Our threshold of 100 citations is in fact more conservative than recent comparable studies (e.g., ref. ¹¹: West and Capellini 2016, Nat Commun; ref.¹²: Dey et al. 2017, Nat Ecol Evol). More importantly, our study incorporates information on the reproductive biology of species whenever available to inform decisions regarding the coding of absence of social grouping, and our use of multiple exclusion criteria (Supp Info: ‘Assigning absence of social grouping’ section) provides supporting evidence for most species”

Of the 20 species assigned absence of social grouping based solely on citations, citation count ranged from 103 – 941 (mean = 260, median =178, see Supplementary Data File 1), suggesting substantial research effort on these species. We acknowledge that no minimum citation threshold guarantees that the behavioural ecology of the species is ‘well studied’, however our approach remains conservative given the methods employed in comparable studies. We have now amended the ‘Literature Searches’ section of the manuscript supp info to contain more of the above justification for our choice of this 100-citation threshold.

Finally, because I am a behavioral ecologist studying social behavior in lizards, I am not strong in the phylogenetic comparative analyses described in the manuscript. I asked and received permission from the journal editor to consult with a departmental colleague, who in fact does these kinds of analyses in his own research. This is what he had to say:

(1) I am not at all familiar with the PGLMMs. I could, however, evaluate the MuSSE and ancestral-state reconstructions. The latter two seemed well done. That said, the authors describe (in the main methods) the PGLMMs as though it were a standard procedure, giving only ONE citation for the whole thing. That gives the impression that PGLMMs are standard, but I’ve honestly not read one single methods paper that introduces them, uses them, or tests their properties. So the authors need to better justify their use of this method and give readers citations to follow for their own use of the methods.

Response: We now provide a citation to the original methods papers underlying these analyses (i.e. ref. ¹³⁻¹⁵, line 338). Accessing the citation lists for these publications yields numerous papers discussing or extending the potential applications of these analyses (e.g. ref. ^{16,17}) as well as many recent examples of studies having utilised this method to conduct phylogenetically controlled tests of character state evolution (e.g. ref. ¹⁸⁻²¹).

(2) They only used one phylogeny. Why just this one? Are the results robust? Also, they say it was the Pyron et al. 2013 phylogeny, but their figures show that they used the time-calibrated phylogeny of Pyron and Burbrink 2014. The two trees have the same topology, but the former has branch lengths in units of genetic substitution. Despite this, the authors cite the former as their “time-calibrated phylogeny” on lines 348–349. So they should cite Pyron and Burbrink 2014 (the real time-calibrated tree) as well.

Response: The phylogeny we used was the time-calibrated tree of Pyron and Burbrink 2014, adapted from the original maximum likelihood phylogeny published by Pyron et al. 2013. With such a large phylogeny (>4000 species), generating an adequate posterior distribution of trees for a fully Bayesian approach was beyond the computational limitations of this study, and assumedly also beyond that of Pyron et al. 2013. Therefore, we chose to rely on this maximum likelihood tree as the best current estimate of phylogenetic relationships among the squamata and confirm the robustness of our results by performing multiple analyses using a range of methodological approaches.

Although this approach limits our ability to incorporate phylogenetic uncertainty into reconstructions, we made efforts to incorporate uncertainty in character state reconstructions themselves by simulating 1000 stochastic character histories for each reconstruction. This allowed us to estimate the error associated with our estimates of transition rates toward social grouping, as well as the number of independent origins of social grouping, from the posterior distribution of parameter estimates from each map set (see lines 354 – 365 of the previous manuscript; ref. ²²).

As for the second point, we thank the reviewer for pointing out this referencing error. We have now amended this line of the manuscript to contain a citation to both publications (line 424).

(3) MuSSE methods and Supp. table 3: In the methods the authors say that they tested their free-parameter model (in which all parameters were free to vary) with “a constrained version of the model in which these transition rates were forced to be equal”. This description is a little misleading, because the only model they test in all these cases is whether a transition TO social grouping is higher in viviparous than oviparous lineages. Is it reasonable to also ask about the LOSS of social grouping? I think it’s reasonable to expect that if viviparity makes it more likely to become social, then oviparity should make it more likely to lose sociality. They don’t test this latter possibility (which would be testing if q21 is higher than q43).

Response: The reviewer raises an important point about the distinction between the maintenance versus origin of social grouping – the conditions that promote the initial emergence of social grouping may not be the same as those that maintain it in the population. For example, once social grouping emerges and stabilises, local ecology may be more important than parity mode in determining whether selection for social grouping relaxes or intensifies. Our test specifically targets the conditions that promote the *origin* of social grouping from a state of no grouping. As such these analyses do not attempt to provide an explanation for the conditions that influence the maintenance or loss of social grouping after its initial evolution. Despite this limitation to inference, we re-fit our models to test whether q12 was higher than q43 and found no evidence to suggest that the rate of loss of sociality differs between parity modes. Furthermore, it is important to note that our power to detect this effect is limited, as there are only 29 oviparous species reported as displaying SG in our data set, and many of the transitions to social grouping in oviparous lineages occur on terminal branches (see figure 3A).

The flip side of this question is to ask about this correlation. That is, could being social lead to viviparity? Maybe that doesn’t make mechanistic sense, but again, if it is possible, they do not test that possibility (i.e. test whether q23 is higher than q14). I would ask them to be more explicit about what they actually did and to justify why they didn’t test these other possibilities (i.e. a biological justification for why testing them does not make sense).

Response: The transition rates q14 and q23 represent simultaneous double transitions in character states (i.e. from oviparity with no social grouping to

viviparity with social grouping and vice versa). These transitions were prohibited in all analyses due to being biologically implausible (see lines 395 – 397; also ref. ²³) and do not precisely address the question posed by the reviewer due to a confound between parity mode and social grouping. However it is possible to test the alternative causation highlighted by the reviewer by testing whether q24 is higher than q13. We can think of no biologically plausible reason why being social might promote the evolution of viviparity, but a re-fit reveals greater support for unconstrained models and that rates of q13 were higher than q24, suggesting that transitions to viviparity from oviparity occur more often in lineages lacking social grouping. These results support our original inference of a causal association between viviparity and the emergence of social grouping. We have amended the supp info by stating these results as an explanation for why we reject these alternative causal explanations.

References

1. Stow, A. J. & Sunnucks, P. High mate and site fidelity in Cunningham's skinks (*Egernia cunninghami*) in natural and fragmented habitat. *Mol. Ecol.* **13**, 419–430 (2004).
2. Davis, A. R., Corl, A., Surget-Groba, Y. & Sinervo, B. Convergent evolution of kin-based sociality in a lizard. *Proc. Biol. Sci.* **278**, 1507–1514 (2011).
3. Gardner, M. G., Bull, C. M., Cooper, S. J. & Duffield, G. A. Genetic evidence for a family structure in stable social aggregations of the Australian lizard *Egernia stokesii*. *Mol. Ecol.* **10**, 175–183 (2001).
4. Chapple, D. G. & Keogh, J. S. Group structure and stability in social aggregations of White's skink, *Egernia whitii*. *Ethology* **112**, 247–257 (2006).
5. Barry, M., Shanas, U. & Brunton, D. H. Year-Round Mixed-Age Shelter Aggregations in Duvaucel's Geckos (*Hoplodactylus duvaucelii*). *Herpetologica* **70**, 395–406 (2014).
6. Shah, B., Shine, R. & Hudson, S. Sociality in lizards: why do thick-tailed geckos (*Nephrurus milii*) aggregate? *Behaviour* **140**, 1039–1052 (2003).
7. Gardner, M. G., Pearson, S. K., Johnston, G. R. & Schwarz, M. P. Group living in squamate reptiles: a review of evidence for stable aggregations. *Biol. Rev. Camb. Philos. Soc.* **91**, 925–936 (2016).
8. Doody, J. S., Burghardt, G. M. & Dinets, V. Breaking the Social-Non-social Dichotomy: A Role for Reptiles in Vertebrate Social Behavior Research? *Ethology* **119**, 95–103 (2013).
9. Freckleton, R. P. The seven deadly sins of comparative analysis. *J. Evol. Biol.* **22**, 1367–1375 (2009).
10. Nakagawa, S. & Freckleton, R. P. Missing inaction: the dangers of ignoring missing data. *Trends Ecol. Evol. (Amst.)* **23**, 592–596 (2008).
11. West, H. E. R. & Capellini, I. Male care and life history traits in mammals. *Nat. Commun.* **7**, 1–10 (2016).
12. Dey, C. J. *et al.* Direct benefits and evolutionary transitions to complex societies. *Nat. ecol. evol.* **1**, 0137–8 (2017).
13. Ives, A. R. & Garland, T. Phylogenetic logistic regression for binary

- dependent variables. *System. Biol.* **59**, 9–26 (2010).
14. Ives, A. R. & Helmus, M. R. Generalized linear mixed models for phylogenetic analyses of community structure. *Ecol. Mon.* **81**, 511–525 (2011).
 15. Ives, A. R. & Garland, T. in *Modern Phylogenetic Comparative Methods and Their Application in Evolutionary Biology* (ed. Garamszegi, L. Z.) 231–261 (Springer Berlin Heidelberg, 2014).
 16. Weber, M. G., Wagner, C. E., Best, R. J., Harmon, L. J. & Matthews, B. Evolution in a Community Context: On Integrating Ecological Interactions and Macroevolution. *Trends Ecol. Evol. (Amst.)* **32**, 291–304 (2017).
 17. Tung Ho, L. S. & Ané, C. A Linear-Time Algorithm for Gaussian and Non-Gaussian Trait Evolution Models. *System. Biol.* **63**, 397–408 (2014).
 18. Arbetman, M. P., Gleiser, G., Morales, C. L., Williams, P. & Aizen, M. A. Global decline of bumblebees is phylogenetically structured and inversely related to species range size and pathogen incidence. *Proc. Biol. Sci.* **284**, 20170204–8 (2017).
 19. Calamari, Z. T. Sexual maturity and shape development in cranial appendages of extant ruminants. *Ecol Evol* **6**, 7820–7830 (2016).
 20. Wagner, C. E., Harmon, L. J. & Seehausen, O. Ecological opportunity and sexual selection together predict adaptive radiation. *Nature* **487**, 366–369 (2012).
 21. Higinson, D. M., Miller, K. B., Segraves, K. A. & Pitnick, S. Female reproductive tract form drives the evolution of complex sperm morphology. *Proc. Natl. Acad. Sci. U.S.A.* **109**, 4538–4543 (2012).
 22. Bollback, J. P. SIMMAP: stochastic character mapping of discrete traits on phylogenies. *BMC Bioinformatics* **7**, 88 (2006).
 23. Lewontin, R. C. *The genetic basis of evolutionary change*. (Columbia University Press, 1974).

REVIEWERS' COMMENTS:

Reviewer #3 (Remarks to the Author):

I have carefully read the most recent response to the reviewers written by the authors and find that the described responses plus those changes they made to the original manuscript version were satisfactory not only for my criticisms, but for those of the other two reviewers, as well. Excellent job by the authors. I will not detail my acceptance of the revisions except to say that the authors did not always use the correct line numbers to reference places in the revised manuscript, but I found those places nevertheless.

I am very satisfied with the revised manuscript and supplementary information.